# High-Throughput Screening of Psychotropic Compounds: Impacts on Swimming Behaviours in *Artemia franciscana*

**DOI:** 10.3390/toxics9030064

**Published:** 2021-03-17

**Authors:** Shanelle A. Kohler, Matthew O. Parker, Alex T. Ford

**Affiliations:** 1Institute of Marine Science Laboratories, Ferry Road, Eastney, Hants PO4 9LY, UK; shanelle.kohler@port.ac.uk; 2School of Pharmacy & Biomedical Science, White Swan Road, St. Michael’s Building, Portsmouth PO1 2DT, UK; matthew.parker@port.ac.uk

**Keywords:** ecotoxicology, behaviour, artemia, psychotropics, behavioural ecotoxicology

## Abstract

Animal behaviour is becoming increasingly popular as an endpoint in ecotoxicology due to its increased sensitivity and speed compared to traditional endpoints. However, the widespread use of animal behaviours in environmental risk assessment is currently hindered by a lack of optimisation and standardisation of behavioural assays for model species. In this study, assays to assess swimming speed were developed for a model crustacean species, the brine shrimp *Artemia franciscana*. Preliminary works were performed to determine optimal arena size for this species, and weather lux used in the experiments had an impact on the animals phototactic response. Swimming speed was significantly lower in the smallest arena, whilst no difference was observed between the two larger arenas, suggesting that the small arena was limiting swimming ability. No significant difference was observed in attraction to light between high and low light intensities. Arena size had a significant impact on phototaxis behaviours. Large arenas resulted in animals spending more time in the light side of the arena compared to medium and small, irrespective of light intensity. The swimming speed assay was then used to expose specimens to a range of psychotropic compounds with varying modes of action. Results indicate that swimming speed provides a valid measure of the impacts of behaviour modulating compounds on *A. franciscana*. The psychotropic compounds tested varied in their impacts on animal behaviour. Fluoxetine resulted in increased swimming speed as has been found in other crustacean species, whilst oxazepam, venlafaxine and amitriptyline had no significant impacts on the behaviours measured. The results from this study suggest a simple, fast, high throughput assay for *A. franciscana* and gains insight on the impacts of a range of psychotropic compounds on the swimming behaviours of a model crustacean species used in ecotoxicology studies.

## 1. Introduction

One of the main challenges facing regulatory risk assessment is the speed with which we can assess the sub-lethal effects of pollutants [1]. Behavioural responses have been indicated as a useful endpoint as they tend to be more sensitive than lethality and faster to assess than endpoints for growth, development and reproduction [2]. However, the use of animal behaviours in environmental risk assessment is currently hindered by a lack of optimisation and standardisation of behavioural assays [2,3]. The use of behavioural hardware mitigates some of the issues of standardisation by providing a controlled environment within which to perform behavioural assays [4]. In recent years, there have been some excellent examples of automated high-throughput behavioural assays with crustacean species. Micro-fluidic behavioural chambers were developed for amphipods (*Allorchestes compressa*) and brine shrimp (*Artemia franciscana*) [5,6,7] and proved to be sensitive assays for measuring alterations in swimming and locomotion in the presence of behaviour modifying compounds. Other studies on zebrafish larvae have successfully used a static, multi-well plate system for high-throughput assessment of compounds on swimming, social, and anxiety behaviours [8,9]. A multi-well plate system is desirable as the plates have standardised dimensions, are readily available, and are compatible with commercial plug-and-play behavioural hardware. In addition to the standardisation of hardware, understanding the baseline unconditioned behaviours of a model species is also important when performing behavioural studies. It has been shown that behaviours can vary with differences in experimental design such as the shape and size of behavioural arenas both between and within species [10,11]. Performing preliminary experiments to understand the baseline behaviours of a model species would help to both optimise the experimental design and aid the interpretation of results from behavioural assays.

Brine shrimp or *Artemia spp* are small crustaceans adapted to hyper-salinity, dry or harsh conditions, and are closely related to other zooplanktons such as the freshwater *Daphnids* [12]. *Artemia spp* have been used as a model species in ecotoxicology testing for more than five decades to assess the potential impacts of environmental pollutants [12,13], and are desirable due to their rapid hatching, cost effectiveness, and commercial availability. *Artemia spp* cysts can be sourced with standardised toxicity kits and hatched under controlled conditions in the lab for fast screening of toxicity in lethality tests (LC50s). Other endpoints, including behaviour, have also proved useful in ecotoxicology testing. The Swimming Speed Alteration (SSA) test was developed by Faimali et al. in 2006 [14] with barnacle larvae, and used video tracking for high-throughput assessment of swimming behaviour. The methods outlined by Faimali et al. have since been applied to *Artemia* by Garaventa et al. in 2010 [15] and Manfra et al. in 2015 [16] who found swimming speed to be more sensitive as an endpoint than mortality.

Psychotropic compounds such as anxiolytics and antidepressants come in a range of different classes with varying modes of action (MOA). In this study, brine shrimp (*Artemia franciscana*) were exposed to environmentally relevant concentrations of fluoxetine hydrochloride, oxazepam, amitriptyline hydrochloride, and venlafaxine hydrochloride; representing the most prescribed compounds from four separate classes of antidepressants and anxiolytics. The MOA, presence in aquatic environments, and effects on animal behaviours are summarised in Table 1.

In the literature, most studies assessing the ecological effects of antidepressants and anxiolytics have focused on fluoxetine. Few have examined the effects of other psychotropic compounds, and fewer still have assessed their effects on aquatic invertebrates. Planktonic crustaceans share with vertebrates several of the neurotransmitters that are targeted by neuroactive drugs. Including serotonin, dopamine, epinephrine and GABA receptors [1], making it possible for psychotropic compounds to have effects on non-target organisms in the environment. From the research to date, there appears to be a trend of increased activity in crustaceans exposed to both anxiolytics and antidepressants including fluoxetine and oxazepam [22,30,43,65] in amphipod and decapod species. To date, the impact of psychotropic compounds on the swimming of anostraca species remains unexplored.

The main aims of the study were to develop a standardised high-throughput behavioural assay for aquatic invertebrates for use in toxicity testing, and to assess the effects of a range of psychotropic compounds with varying modes of action on crustacean behaviour. Behavioural assays were developed for *A. franciscana* and data on the baseline unconditioned swimming behaviours were collected. The swimming speed and photosensitivity was assessed under a range of arena sizes. It was thought that as has been shown in amphipods [10], that smaller arenas would limit swimming speeds. Some studies have reported that both adult and larval *Artemia spp* can switch between positive and negative phototaxis depending on the intensity of light used [66,67]. As a result of this, the phototactic response of *A. franciscana* was also assessed under different light intensities. It was hypothesised that at higher light intensities *A. franciscana* would exhibit a preference for dark areas which would be reduced or mitigated at lower intensities. Following assay development, *A. franciscana* were exposed to three antidepressants and an anxiolytic at environmentally relevant concentrations and swimming behaviours were assessed. Based on the current literature for crustaceans, it was hypothesised that psychotropic compounds would increase swimming speed in *A. franciscana*.

## 2. Materials and Methods

### 2.1. Animal Culture and Husbandry

*A. franciscana* were purchased from MicroBioTests Inc (Kleimoer 15 9030 Gent, Belgium), as dried cysts and hatched in a 1 L separating funnel connected to an air pump. Following hatching, organisms were transferred to a 5 L aquarium with an air stone. The hatchery and aquaria were set up within an incubator to keep temperature and light conditions consistent. Cool white, fluorescent lamps were used, and light intensity ranged between 1665–1608 Lux (21.73–22.51 µmol·s^−1^·m^−2^) from the top to the bottom of the incubator, respectively. Hatching and growth parameters were in accordance with the MicroBioTests Artoxkit M protocol. Artificial seawater (AFSW) was used at 35 ppt and a constant temperature of 21 ± 1 °C. A 12:12 light/dark regime was used during both hatching and growth of organisms. Once nauplii were transferred from the separating funnel to the growing aquaria, a water change was performed every 3 days, and animals were fed 2–4 drops of concentrated algae solution containing *Nannochloropsis spp* and *Tetraseimis spp* (purchased from Amazon.co.uk by supplier Phyto Plus) every 1–2 days. The amount of food added was judged by eye based on the colour of the aquarium water, as per instructions on the algae solution bottle, to obtain a light green tint to the culture water. Nauplii were kept in the growing aquaria and reared to adult stage. It took between 3–4 weeks to rear *A. franciscana* from Instar stage I to trackable sized adults with a mean body length of 10 mm.

### 2.2. Measuring Behaviour

All behaviours were measured using DanioVision^™^ observation chamber (Noldus, Wageningen, The Netherlands) connected to EthoVision^®^XT 11.5 software (TrackSys, Nottingham, UK). The observation chamber was comprised of an external hood and internal holder for a multi-well plate. The holder is infrared backlit with an additional cold white light source which can be programmed to operate automatically. Together these provide a controlled environment for behavioural experiments. The EthoVision^®^XT 11.5 software can measure a variety of parameters associated with movement and activity simultaneously and can be programmed to operate DanioVision^™^ hardware.

### 2.3. Baseline Behaviours

Prior to psychotropic exposures, preliminary tests were performed to determine the optimal arena size for behavioural assays. Standard Thermo Scientific ‘Nunc’ 6-well, 12-well and 24-well plates (sourced from Thermo Fisher Scientific, Waltham, MA, USA) were used to measure the baseline unconditioned behaviours of *A. franciscana*.

### 2.4. Velocity

To measure swimming speed, animals were gently transferred from growth tanks and loaded into multi-well plates using a plastic Pasteur pipette. The pipette was widened by cutting the tip so that *A. franciscana* could be transferred without physical damage. A single individual was placed in each well. For ease of writing, the 24-well, 12-well and 6-well plate will be henceforth referred to as small, medium, and large arenas, respectively. Each well was filled to half of its maximum volume with AFSW which allowed for free horizontal swimming but limited vertical motions. The dimensions of arenas including the volume of AFSW used and the number of replicates of *A. franciscana* are outlined in Table 2.

Once loaded, multi-well plates were placed inside the DanioVision^™^ and animals were tracked for 8 min under a 2 min dark: 2 min light, cycle. The differing light phases were used as a stimuli and to assess the photosensitivity of *A. franciscana*. The cold light was set to 100% intensity equating to 4000 Lux. This was almost double the Lux used for hatching and culturing of organisms to try and combat habituation to the lighting in the behaviour trials.

#### Phototaxis

The small, medium and large arenas were also used to assess the effects of arena size on the baseline unconditioned phototaxis behaviour in *A. franciscana*. Animals were tracked in the DanioVision™ for 6 min with a 3 min dark phase followed by a 3 min light phase. Here, the light phase was used to measure a phototactic response. A series of custom acrylic strips were used. The strips consisted of a clear acrylic that both white light and infra-red light could pass through and a black acrylic through which only the infra-red light could pass. During the dark phase the entire arena was dark and during the light phase one half of the arena was illuminated whilst the other half remained dark and *A. franciscana* could choose to be in either the light or dark side of the arena. Zone use during the dark phase when the entire arena was dark was used to control for animals that generally preferred one side of an arena compared to another. It was expected that during dark phases animals would use all of the arena equally. During light phases, animals would exhibit phototaxis if they then showed a preference for either the light or dark side of the arena. The acrylic strips were produced in a range of sizes to provide a half-light and half-dark side of the arena for each size class, the dimensions of the light and dark zones within each arena are outlined in (Figure 1). Two plates were made for each size arena which could be interchanged during trials so that the light and dark side of the arenas could be alternated at random (Figure 1).

To assess the impacts of light intensity on phototactic response of *A. franciscana.* The light phase for phototaxis trials were performed under two light intensities. Two conditions 5% and 100% light intensity (200 and 4000 Lux, respectively) were used. A total of 312 animals were used for phototaxis assessment with replicates divided between arena size, acrylic plate, and light intensity. The number of replicates used for each condition are outlined in Table 3.

### 2.5. Psychotropic Exposures

#### Preparation of Solutions

Following preliminary experiments, *A. franciscana* were exposed to a range of psychotropic compounds. All compounds were sourced from Sigma-Aldrich (Saint Louis, Missouri, USA) in dry powder form including Fluoxetine hydrochloride (CAS: 56296-78-7), Oxazepam (CAS: 604-75-1), Amitriptyline hydrochloride (CAS: 549-18-8), and Venlafaxine hydrochloride (CAS: 99300-78-4). All compounds were water soluble, so solutions were made without a solvent. Due to the minimum amount of dry compound that can be accurately weighed, a stock solution of 1 mg/L was made in 2 L volumetrics for each compound. Stock solutions were stored in sealed glass vials wrapped in aluminium foil and stored in the fridge at 10 ± 1 °C to prevent degradation. A serial dilution of 10 ng/L, 100 ng/L and 100 ng/L plus an AFSW control was made for each compound, from stock solutions, into artificial seawater at 35 ppt.

### 2.6. Exposures and Behavioural Analysis

The experimental design for psychotropic exposures is outlined in Figure 2. Medium arenas were used, as per results from the studies on baseline behaviours, as this provided the best trade-off between high-throughput analysis and providing ‘space to behave’ in this species. A single individual of *A. franciscana* was loaded into each well with water from the culture tanks. Once all animals were in the arenas, the culture water was removed from the wells with an electronic pipette and replaced with 4 mL of AFSW control or AFSW spiked with a psychotropic compound at 10 ng/L, 100 ng/L or 1000 ng/L. 12 replicates were performed per concentration providing a total of 48 animals per compound. The organisms were then placed in an incubator under a 12:12 h light:dark cycle at 21 °C ± 1. After 1 h, the well plates were removed from the incubator and placed in the DavioVision^™^. *A. franciscana* were tracked using EthoVision^®^XT software for a total of 8 min under a 2 min dark: 2 min light cycle. The light phase was set to 100% light intensity (4000 Lux) as per results from studies on baseline behaviours. The process was repeated with the same animals after 1 day and 1 week of exposure.

### 2.7. Statistics

When analysing baseline behaviours, statistical analysis was performed in IBM SPSS Statistics 24. Phototaxis was measured as the percent duration of time spent in the light zone of the arena. Total distance moved was included in the model as a co-variate to correct for animals that did not move during trials [3]. Velocity was measured as mean velocity in centimetres per second and was analysed in both 2-min and 10-s time bins. Extreme anomalous values generated by the loss of tracking by the EthoVision^®^XT software was excluded from the data analysis (as defined by values > median ± 3*IQR) and never removed more than 3% of data points. Linear mixed effects (LME) models were used for all comparisons, residuals from LME model analysis were checked for normality using Q-Q plots and Shapiro-Wilk test of normality. All datasets were normally distributed. Treatment, length of exposure and time were input as factors. Individual animal ID was included as a random effect in the model to correct for repeated measures. Post Hoc analysis was performed using Bonferroni adjustments to correct for type- II errors. *p*-values of < 0.05 were considered significant. When analysing data from the psychotropic exposures, statistical analysis was performed with jamovi version 1.2.27. Velocity was analysed in both 2-min and 10-s time bins. Linear mixed effects models were used for all compounds. For the 10-s data treatment and length of exposure were input as factors whilst time was used as a covariate. Individual ID was used as a random effect in the model to correct for repeated measures. Post Hoc analysis was performed using Holmes adjustments to correct for type-II errors. For the 2-s data, treatment, length of exposure and time in the form of 2-min light or 2-min dark phases were all input as factors. Individual ID was used as a random effect in the model to correct for repeated measures. Post Hoc analysis was performed using Holmes adjustments. No differences were observed in the data output from the mixed effects models between 2-min and 10-s data. As such only the 2-min data has been presented in the results. All figures were made using the estimated marginal means from linear mixed effects models.

## 3. Results

### 3.1. Baseline Behaviours

#### Velocity

When assessing the baseline unconditioned velocity behaviour of *A. franciscana*, no statistical differences were observed between the 2-min and 10-s time bins when comparing arena sizes or light phase (Table 4). A significant effect of arena size in both 2-min and 10-s time bins was observed (Table 4). *A. franciscana* reached a greater mean velocity, in a range of 1.1–1.5 cm/s in large and medium arenas compared to small arenas where animals reached a maximum velocity of 0.9 cm/s (Figure 3). No significant effects (*p* > 0.05) were observed in velocity between large and medium arenas for 2-min time bins (Figure 3A). A significant effect of time was observed with *A. franciscana* swimming faster during dark phases compared to light (Table 4). A significant interaction was observed between arena size and time when splitting data into 10-s time bins but not with 2-min time bins (Table 4). The significant interaction was driven by *A. franciscana* swimming faster in the large arena compared to medium and small during the second dark phase (Figure 3B).

### 3.2. Phototaxis

When assessing the baseline unconditioned phototactic behaviour of *A. franciscana*, arena size had a significant impact on time spent in the light zone (Table 5). Animals spent significantly more time in the light zone when in large arenas compared to medium and small. Light phase also had a significant impact on phototactic response (Table 5) with animals spending more time in the light zone during light phases compared to dark. No significant effects were observed in phototactic response between the two light intensities (Table 5). There was a significant interaction between arena size and time (Table 5). There were no observed differences between arena sizes during 3-min dark phases with all animals spending ~50% of their time in the light side of the arena (Figure 4). However, during light phases *A. franciscana* spent a greater proportion (~55%) of time in the light zone when in the large arena and a smaller proportion of time (~35–50%) in the light zone when in a medium or small arena (Figure 4).

### 3.3. Psychotropic Exposures

#### Fluoxetine

When assessing the effects of fluoxetine on the velocity of *A. franciscana*, mean velocity ranged between 0.3–0.7 cm/s across all treatments and exposures (Figure 5D–F). Fluoxetine had a significant impact on swimming speed between treatments (Table 6). Animals exposed to 100 ng/L of fluoxetine had a significantly greater velocity than both control animals and those exposed to the lowest treatment group of 10 ng/L (Figure 5A). Animals in the 1000 ng/L treatment group also reached a greater mean velocity than controls and animals exposed to 10 ng/L, but this did not reach the threshold for significance following multiple testing correction (*p* = 0.072). Light phase had a significant effect on the velocity of *A. franciscana* (Table 6) with animals generally swimming faster during dark phases compared to light (Figure 5B). The length of exposure also had a significant effect on swimming speed (Table 6) with animals swimming significantly slower time (Figure 5C). No significant interactions were found between fluoxetine treatments with length of exposure or light phase (Table 6; Figure 5D–F).

### 3.4. Oxazepam

For the oxazepam study, the mean velocity of *A. franciscana* ranged between 0.3–0.7 cm/s (Figure 6D–F). No significant effects of oxazepam were observed between treatments (Table 7; Figure 6A). Velocity was significantly different between light phases (Table 7) with animals swimming faster during the second dark phase compared to the two light phases (Figure 6B). Mean velocity significantly decreased with increasing of length of exposure (Table 7; Figure 6C), and a significant interaction was observed between light phase and exposure (Table 7). The significant interaction was driven by the 1-h exposure, in which velocity was significantly greater during the two dark periods compared to the two light periods, and significantly greater than 1 day and 1-week exposures (Table 1). No significant interactions were found between oxazepam treatments with length of exposure or light phase (Table 7; Figure 6D–F).

### 3.5. Amitriptyline

During the amitriptyline study, the swimming velocity of *A. franciscana* ranged between 0.3–0.65 cm/s (Figure 7D–F). Light phase had a significant effect on swimming behaviours (Table 8) with *A. franciscana* reaching a greater velocity during the second dark phase compared to the first and second light phase (Figure 7B). Length of exposure also had a significant effect on velocity (Table 8) with swimming speed decreasing significantly after 1 week of exposure compared to 1 h (Figure 7C). Whilst animals exposed to the highest concentrations of amitriptyline swam slower than the controls after 1 h and 1 day of exposure (Figure 7D,E), this did not meet the significance threshold. No significant effects of amitriptyline treatment were observed on Artemia velocity (Table 8; Figure 7A) and no significant interactions were observed between treatment with light phase and/or exposure (Table 8; Figure 7D–F).

### 3.6. Venlafaxine

When assessing the effects of venlafaxine on the velocity of *A. franciscana*, swimming velocity ranged between 0.3–0.6 cm/s (Figure 8D–F). Light phase had a significant effect on swimming behaviours (Table 9) with animals swimming faster during dark phases compared to light (Figure 8B). Length of exposure also had a significant effect on velocity (Table 9) with swimming speed decreasing significantly after 1 week of exposure compared to 1 day (Figure 8C). It was observed that animals in the highest treatment group swam consistently slower than control animals across all three measured durations of exposure (Figure 8D–F). However, this did not reach a level of significance and no significant impacts of venlafaxine were observed on the velocity of *A. franciscana* between treatment groups (Table 9; Figure 8A). No significant interactions were observed between treatments with light phase and/or length of exposure (Table 9; Figure 8D–F).

## 4. Discussion

### 4.1. Baseline Behaviours

There is growing evidence that collection of baseline data on model species is important when conducting behavioural assays. Researchers have highlighted a lack of baseline data to be a source of variability in the results of behavioural studies in ecotoxicology [68]. This has been supported by Melvin et al. [69] who explored how fish acclimate to behavioural arenas and how different lengths of observation time impact estimates of basic swimming parameters. They concluded that researchers need to establish a basic knowledge about the baseline behavioural characteristics for a model species, as this could influence study outcomes of behavioural ecotoxicology experiments. This was further reinforced by Kohler et al. [10,11] who discovered that differences in study design could impact the baseline unconditioned behaviours in amphipods which could in turn implicate the results of ecotoxicology studies.

In this study, experiments were performed to assess the baseline unconditioned velocity and phototaxis behaviours of *A. franciscana* under a range of arena sizes to both optimise the assay for high-throughput analysis and to determine the sensitivity of the test species to the behavioural assays. Results from velocity assessments indicate a trade-off between ‘high-throughput’ analysis and providing ‘space to behave’ with *A. franciscana* reaching a significantly greater velocity in large and medium arenas compared to small suggesting that the small arenas were limiting animals ability to reach a greater swimming speed. Analysing velocity data in 2-min time bins found no significant differences in swimming speed between the large and medium arenas, whereas the 10-s analysis found that animals reached a significantly faster swimming speed in large arenas compared to medium in the second dark phase. Similar results were also described by Kohler et al. [11] in amphipods whereby differences in swimming behaviours between a marine and freshwater amphipod were only observed when data was separated into smaller time bins. With the exception of the 30 s in the second dark phase, no significant differences were observed in swimming speed between medium and large arenas in an 8-min behaviour trial. This suggests that arena size was no longer a limiting factor on *Artemia* velocity and that any further increase in arena size would no longer impact the swimming speed that this species could reach. Increased swimming speed with increasing space to explore has been reported in a range of vertebrate and invertebrate species including the amphipod *G. pulex* [10] fruit flies *Drosophila melanogaster* [70]; rats [71]; and gerbils [72].

During swimming speed assays, light phase had a significant impact on the velocity of *A. franciscana*. Swimming speed was greater during 2-min dark phases compared to light. A transition into darkness may trigger migration behaviours or increase exploratory behaviour as a result of perceived reduction in predation risk or increased searching for light areas. In the literature it has been reported that many zooplankton, including brine shrimp, undergo nocturnal diel vertical migration (DVM) involving an ascent in the water column to feed during times of low light levels near the surface and descend to dim lit areas during the day to avoid predators [73,74]. It has been reported that *Artemia spp* can switch between positive and negative phototaxis depending on the intensity of light used [66,67]. However, in this study, no significant differences in time spent in the light zone were observed between the 2000 and 400 Lux phototaxis trials in this study. No effects of Lux on crustacean behaviours has also been reported previously in the literature for amphipod species [10]. In the study by Kohler et al. in 2018 [10], no significant effect of varying lux were observed for multiple behavioural endpoints including swimming speed and thigmotaxis behaviours. This suggests that the switch between positive and negative phototaxis which was observed by Bradley et al. (1984) and Dojmi Di Delupis et al. (1988) [66,67] was a result of something more complex than simple lux in the control of *Artemia* migration patterns and provides and area for future research. In phototaxis trials, under both light intensities, animals in large arenas exhibited a preference for the light side of the arena compared to the dark. This would support the theory of increased light searching to explain the greater velocity observed during dark phases in swimming speed assays. Interestingly the opposite was observed when assessing phototaxis behaviours in the medium and small arenas. However, this may be the result of the arenas small size resulting in bleeding of light into the dark zone from the bright half of the arena. An increase in activity when in the dark has also been observed in zebrafish. It has been observed that a sudden transition to darkness results in a significant and sudden increase in locomotor activities which have been described as a light searching behaviour and is attributed to increased stress or anxiety [75,76,77]. This theory may also be the case for *A. franciscana*. However, it is worth noting that the increase in activity during dark phases was neither immediate upon sudden transition to the dark, nor consistent for every dark phase which would be expected for a startle or escape response associated with anxiety or stress.

### 4.2. Psychotropic Exposures

Here, *A. franciscana* were exposed to environmentally relevant concentrations of four psychotropic compounds with varying modes of action, and their swimming behaviours were assessed. Fluoxetine had a significant impact on the swimming speed of *A. franciscana* between treatment groups in that velocity was greater in the two highest treatment groups compared to the lowest treatment and control animals. Pairwise comparisons revealed that velocity in the 100 ng/L treated animals was significantly greater than the controls and the 10 ng/L treated animals. There are many studies in the literature which have reported increased activity when exposed to fluoxetine in crustaceans including amphipods [22,30]; shore crabs [65]; crayfish [39]; and *Daphnia* [78]. The results from this study appear to support the current literature and our hypothesis that fluoxetine would increase activity in *A. franciscana*.

No significant impacts on swimming behaviours were observed for oxazepam amitriptyline or venlafaxine. It was noted that both amitriptyline and venlafaxine exposure consistently reduced swimming speeds compared to controls, albeit these results failed to pass our significance threshold. Whether these represent real effects would need further investigation however effect sizes revealed through the ICC and R^2^ were modest at best. It was, however, interesting to note that both of these compounds target norepinephrine receptors. Both norepinephrine and octopamine have been suggested to reduce swimmeret rhythm in crayfish [79] although the exact role of NE in crustaceans is still debated [80].

It has been reported in the literature that compounds with varying modes of action can result in significant effects being found for some compounds, while others have no impacts on behaviour. A study on crayfish reported a significant increase in activity following oxazepam exposure while no significant impacts were observed for venlafaxine [43]. Studies on *Daphnia* behaviour with multiple psychotropic compounds found that fluoxetine induced the most severe behavioural effects and impacted behaviours at the lowest concentrations compared to all other compounds tested [1,81]. It was hypothesised that all of the psychotropic compounds tested in this study could impact the behaviour of *A. franciscana* as they share many of the neurotransmitters that are targeted by neuroactive drugs in vertebrates. This hypothesis was not supported by the results found in this study which instead suggest that oxazepam, amitriptyline, and venlafaxine have no significant effects on swimming speed. This is not to say that these compounds could not influence other behaviours. It has been reported that compounds can affect organisms independently of their intended pathway. In a recent study by Rivetti et al. in 2018 [82], the genes encoding for serotonin synthesis were deleted in *D. magna* generating mutants completely deprived of serotonin. Fluoxetine altered behavioural responses in wild type *D. magna* that had serotonin but had no effect on serotonin deprived mutants as expected for compounds acting via the serotonergic-pathway. However, fluoxetine impacted fecundity and life-history responses of both mutants and wild type *D. magna* suggesting that this drug affects reproduction independently of the serotonin pathway. It has also been reported that psychotropic compounds can impact some behaviours but not others. Tierney et al. [39] found that fluoxetine significantly reduced locomotion in juvenile crayfish but had no impact on thigmotaxis or sheltering behaviours. A study by Mesquita et al. in 2011 [65] reported that crabs exposed to fluoxetine were significantly more active than unexposed crabs, but no differences were observed in their speed. Whilst this study found no significant effects of oxazepam, amitriptyline, and venlafaxine on swimming speed, experiments on other endpoints would be required to elucidate whether it is just swimming speed or whether *A. franciscana* are not sensitive to these compounds.

In this study, the length of exposure significantly impacted the swimming speed of *A. franciscana* across all experiments, independently of compound or dose, with animals swimming slower with increased length of exposure. This could be explained by the static nature of the test arena used for compound exposure. Previous studies on *A. franciscana* found that immobility and fatality of *Artemia* larvae significantly increased after 12 h in a static system, whereas under constant water flow in a microfluidic system, activity levels remained unchanged after 18 h [6]. The reduction in artemia health and mobility was attributed to a depletion of oxygen in a static system. Medium arenas were used for psychotropic exposures in this study as per results from preliminary experiments. The medium arena size allowed for a greater volume of water and surface area compared to small arenas, and a water change was performed after three days to combat the effects of oxygen depletion and compound degradation. Mortality was 12.5% after 1 week of exposure and 29% of deceased animals came from the control groups suggesting that mortality and the decreased activity of *A. franciscana* throughout the experiment was more likely a result of oxygen depletion rather than effects of toxicity from the compounds. It is also possible that the reduction of activity of *A. franciscana* was the result of habituation to the behavioural system. Habituation to behavioural assays has also been reported in a wide range of both vertebrate and invertebrate species [39,70,83,84,85] and has been demonstrated in *E. marinus* and *G. pulex* [10,11]. Repeating the experiment under flow through conditions may help to elucidate weather the reduction in swimming speed observed in *A. franciscana* was a result of depleted oxygen or habituation to the behavioural assay.

## 5. Conclusions

In this study baseline data was collected on swimming and phototactic behaviours of A. franciscana. A trade-off was observed between high-throughput analysis and providing space to behave. Results indicate a 12-well was the minimum arena size in which swimming behaviours were not limited. Arena size also had a significant impact on phototactic behaviours, but light intensity did not. Analysing velocity data in 10-s time bins found differences between medium and large arena sizes whereas 2-min time bins did not suggesting that the increased sensitivity of the 10 s time bin may be necessary for elucidating subtle differences between treatments. Following the collection of baseline data, behavioural assays to assess swimming speed and photosensitivity were developed. Velocity proved a useful endpoint to measure swimming speed and photosensitivity in *A. franciscana*. Compounds with differing MOAs varied in their impacts on animal behaviours. The results from this study further support the evidence that fluoxetine can impact swimming behaviours at environmentally relevant concentrations. Interestingly, no significant effects were observed in the other compounds although it was noted that those that target norepinephrine as well as serotonin displayed reduced swimming activity. We suggest a simple, fast, high throughput assay for *A. franciscana* and provides a baseline on the impacts of a range of psychotropic compounds on the swimming behaviours of a model crustacean species used in ecotoxicology studies.

## Figures and Tables

**Figure 1 toxics-09-00064-f001:**
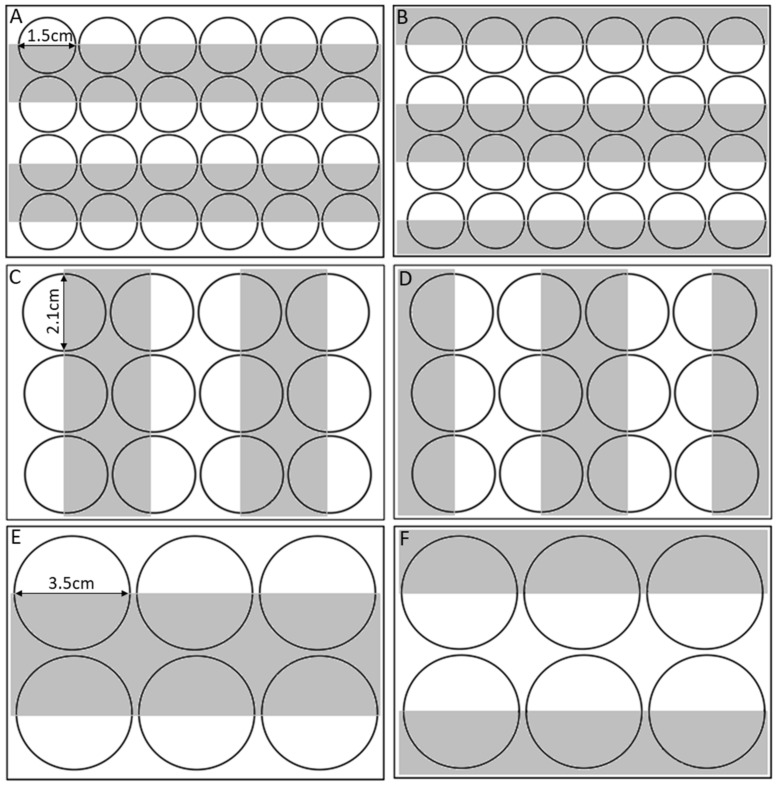
Dimensions and location of light and dark zones for (**A**) small arena, acrylic plate 1, (**B**) small arena, acrylic plate 2, (**C**) medium arena, acrylic plate 1, (**D**) medium arena, acrylic plate 2, (**E**) large arena, acrylic plate 1, (**F**) large arena, acrylic plate 2.

**Figure 2 toxics-09-00064-f002:**
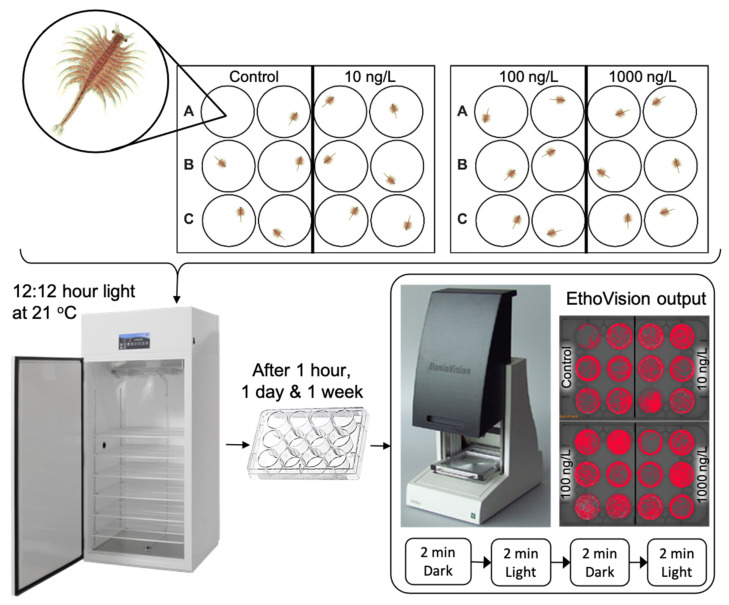
Experimental design for *A. franciscana* exposure and behavioural analysis. The 12-well plates were loaded in duplicate to provide 12 replicates per treatment. The procedure was repeated for each of the four compounds.

**Figure 3 toxics-09-00064-f003:**
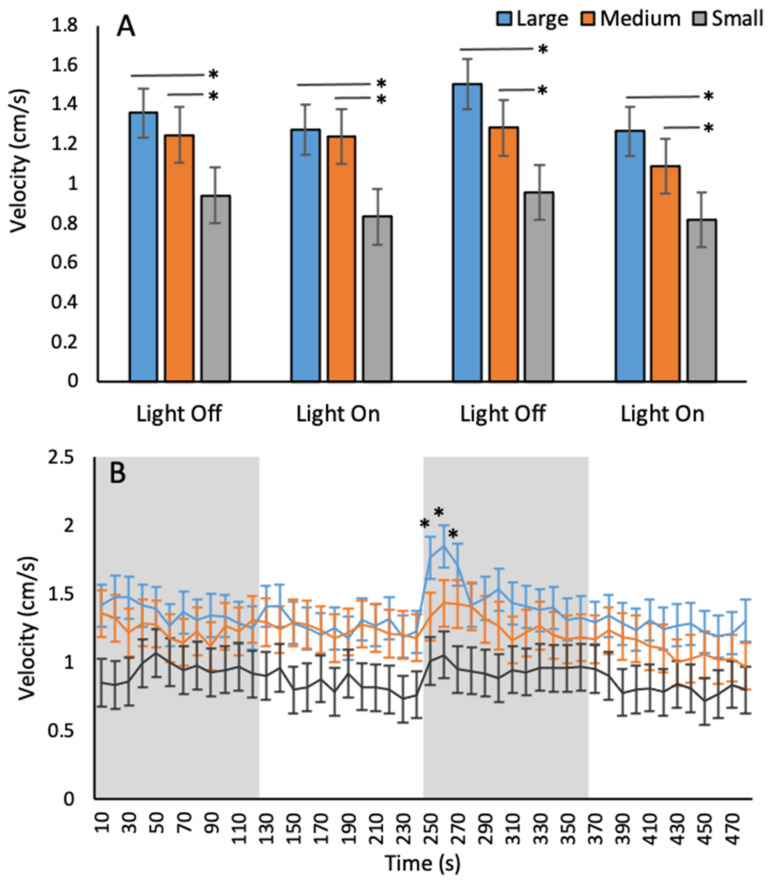
Mean velocity of *A. franciscana* between arena sizes in (**A**) 2-min and (**B**) 10-s time bins. Error bars represent 95% confidence. Asterisks indicate significant differences between arena sizes. For the 10-s data, asterisks indicate significant differences in velocity between large and medium arena only. Significance level * *p* ≤ 0.05.

**Figure 4 toxics-09-00064-f004:**
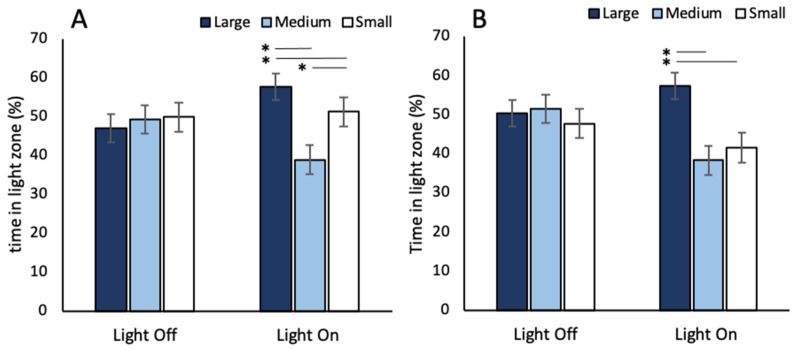
Percent duration *A. franciscana* spent in the light zone during 3-min dark and 3-min light phases between small, medium and large arenas when exposed to light at (**A**) 100% intensity and (**B**) 5% intensity. Error bars indicate 95% confidence. Asterisks indicate significant differences between arena sizes. Significance level * *p* ≤ 0.05.

**Figure 5 toxics-09-00064-f005:**
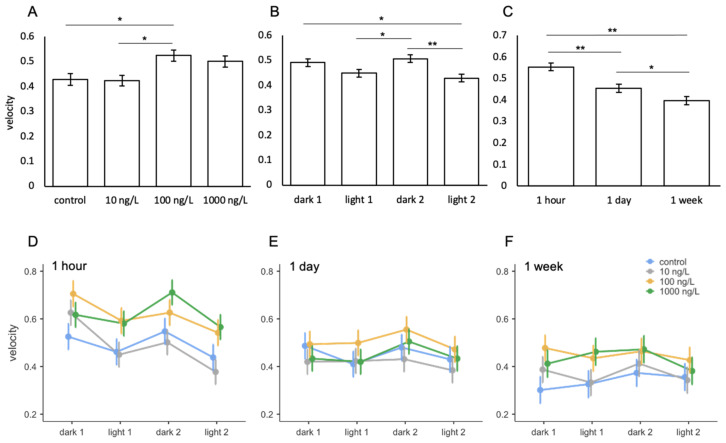
Mean velocity of *A. franciscana* following exposure to fluoxetine between (**A**) treatment, (**B**) light phase, (**C**) length of exposure, and (**D**–**F**) interactions between treatment and light phase across the three lengths of exposure. Error bars represent standard error. Asterisks indicate significant differences from post hoc analysis. * *p* ≤ 0.05, ** *p* ≤ 0.001.

**Figure 6 toxics-09-00064-f006:**
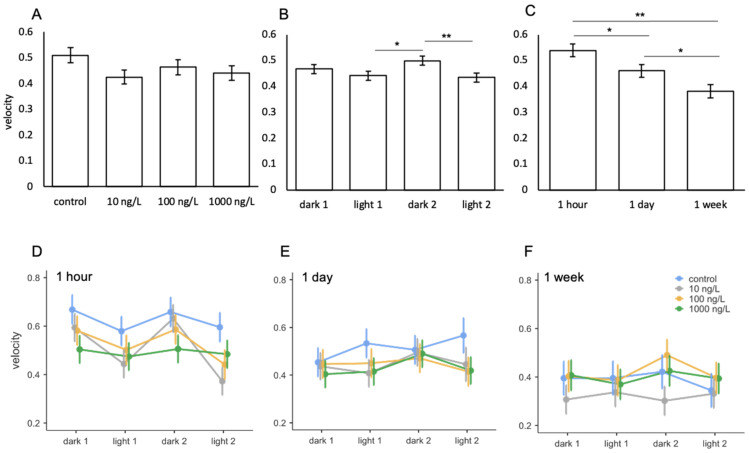
Mean velocity of *A.*
*franciscana* following exposure to oxazepam between (**A**) treatment, (**B**) light phase, (**C**) length of exposure, and (**D**–**F**) interactions between treatment and light phase across the three lengths of exposure. Error bars represent standard error. Asterisks indicate significant differences from post hoc analysis. * *p* ≤ 0.05, ** *p* ≤ 0.001.

**Figure 7 toxics-09-00064-f007:**
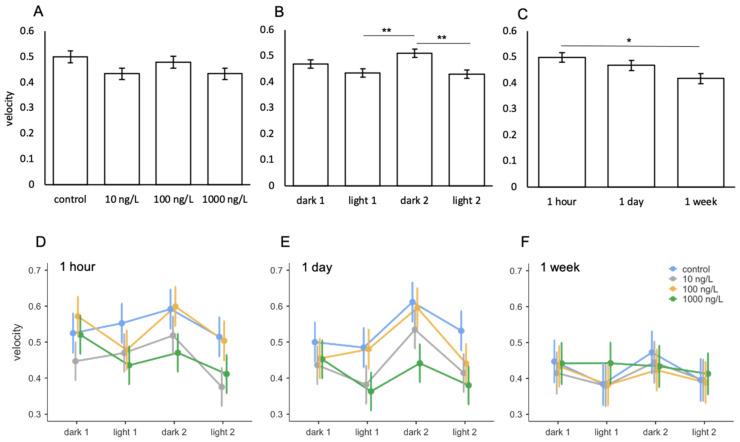
Mean velocity of *A. franciscana* following exposure to amitriptyline between (**A**) treatment, (**B**) light phase, (**C**) length of exposure, and (**D**–**F**) interactions between treatment and light phase across the three lengths of exposure. Error bars represent standard error. Asterisks indicate significant differences from post hoc analysis. * *p* ≤ 0.05, ** *p* ≤ 0.001.

**Figure 8 toxics-09-00064-f008:**
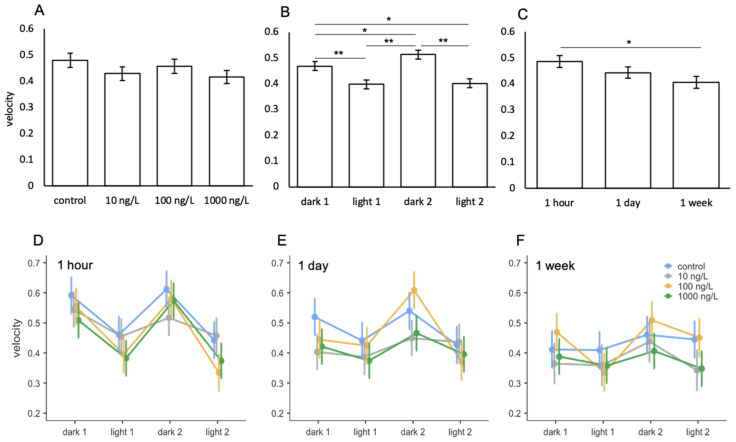
Mean velocity of *A. franciscana* following exposure to venlafaxine between (**A**) treatment, (**B**) light phase, (**C**) length of exposure, and (**D**–**F**) interactions between treatment and light phase across the three lengths of exposure. Error bars represent standard error. Asterisks indicate significant differences from post hoc analysis. * *p* ≤ 0.05, ** *p* ≤ 0.001.

**Table 1 toxics-09-00064-t001:** Summary of the current literature for the four psychotropic compounds used in this study including their MOA, presence in aquatic environments, and effects on behaviours.

Compound	Class	Environment	Concentration	Source	Behaviour Impacts	Source
fluoxetine hydrochloride	Selective Serotonin Reuptake Inhibitor (SSRI)	effluents	0.001–5 µg/L	[17,18,19,20]	activity	[21,22]
		surface waters	0.012–0.02 µg/L		reproduction	[23,24,25,26,27]
		marine env	0.012 µg/L		aggression	[27,28,29]
					feeding	[30,31,32]
					predator avoidance	[21,32,33,34]
					stress/anxiety	[35,36,37]
					taxis	[38,39]
oxazepam	Benzodiazepine (BZD)	effluents	0.25–0.73 ug/L	[18,20,40]	activity	[41,42,43]
		surface waters	0.02–0.58 ug/L		feeding	[44]
					boldness	[41,45,46]
					social behaviour	[44]
					migration	[47]
amitriptyline hydrochloride	Tricyclic Antidepressant (TCA)	effluents	<2–357 ng/L	[48,49,50,51]	activity	[52]
		surface waters	<0.5–72 ng/L		reproduction	[53]
		bio-solids	263–632 ng/g		feeding	[54]
					stress/anxiety	[55]
					memory & learning	[56]
					orientation	[57]
venlafaxine hydrochloride	Selective Serotonin and Norepinephrine Reuptake Inhibitor (SNRI)	effluents	600–1454 ng/L	[49,50]	activity	[58,59]
		surface waters	187 ng/L		feeding	[60]
		bio-solids	289–499 ng/g		stress/anxiety	[61,62,63,64]

**Table 2 toxics-09-00064-t002:** Dimensions of Small (24-well), medium (12-well) and large (6-well) arenas with volume of AFSW and number of replicates of *A. franciscana* used for velocity studies.

Arena	Diameter	Area	Volume AFSW	Replicates
Small	1.5 cm	3 cm^3^	1.5 mL	24
Medium	2.1 cm	6 cm^3^	3 mL	24
Large	3.5 cm	16.4 cm^3^	8 mL	30

**Table 3 toxics-09-00064-t003:** Number of replicates used for each experimental condition for assessment of phototaxis in *A. franciscana* for each of the three arena sizes and the two acrylic plates to alternate the light and dark zones between the two light intensities.

Arena Size	Acrylic Plate Used	Light Intensity	Replicates
Small	1	100%	24
		5%	24
	2	100%	24
		5%	24
Medium	1	100%	24
		5%	24
	2	100%	24
		5%	24
Large	1	100%	30
		5%	30
	2	100%	30
		5%	30

**Table 4 toxics-09-00064-t004:** Output from linear mixed effects model for both 2-min and 10 s velocity data of *A. franciscana* between arena sizes. Significance level *p* < 0.05. In this model ‘time’ represents light phase split into 10 s time bins or 2-min time bins.

Comparison	2-min				10-s			
	N-df	D-df	F	*p*	N-df	D-df	F	*p*
arena size	2	75	16.3	<0.001	2	74	16.2	<0.001
time	3	225	13.6	<0.001	47	3522	7.0	<0.001
arena size * time	6	225	1.5	0.180	94	3522	1.7	<0.001
ICC	0.679				0.475			
R^2^	0.271				0.217			

**Table 5 toxics-09-00064-t005:** Output from linear mixed effects model for time spent in the light side of the arena for *A. franciscana* between arena sizes and light intensity. Significance level *p* < 0.05.

Comparison	Num df	Den df	F	*p*
arena size	2	344.64	13.37	<0.001
light intensity	1	302.92	1.02	0.313
light phase	1	337.59	4.35	0.038
arena size * light intensity	2	303.77	3.72	0.025
arena size * light phase	2	305.09	54.75	<0.001
light intensity * light phase	1	303.25	8.07	0.005
arena size * light intensity * light phase	2	307.19	0.73	0.483
ICC = 0.364				
R^2^ = 0.198				

**Table 6 toxics-09-00064-t006:** Output from linear mixed effects model for velocity of *A. franciscana* exposed to fluoxetine.

Comparison	F	Num df	Den df	*p*
treatment	5.328	3	111	0.002
light phase	8.221	3	392	<0.001
exposure period	17.400	2	111	<0.001
treatment * light phase	0.757	9	392	0.657
treatment * exposure period	0.663	6	111	0.679
light phase * exposure period	1.728	6	392	0.113
treatment * light phase * exposure period	0.520	18	392	0.949
ICC = 0.315				
R^2^ = 0.212				

**Table 7 toxics-09-00064-t007:** Output from linear mixed effects model for velocity of *A. franciscana* exposed to oxazepam.

Comparison	F	Num df	Den df	*p*
treatment	1.602	3	114	0.193
light phase	6.490	3	367	<0.001
exposure period	9.926	2	114	<0.001
treatment * light phase	0.563	9	367	0.827
treatment * exposure period	0.534	6	114	0.782
light phase * exposure period	2.783	6	367	0.012
treatment * light phase * exposure period	1.109	18	367	0.341
ICC = 0.557				
R^2^ = 0.174				

**Table 8 toxics-09-00064-t008:** Output from linear mixed effects model for velocity of *A. franciscana* exposed to amitriptyline.

Comparison	F	Num df	Den df	*p*
treatment	2.204	3	112	0.092
light phase	8.415	3	386	<0.001
exposure period	4.258	2	112	0.017
treatment * light phase	0.537	9	386	0.847
treatment * exposure period	0.703	6	112	0.648
light phase * exposure period	0.861	6	386	0.524
treatment * light phase * exposure period	0.393	18	386	0.989
ICC = 0.318				
R^2^ = 0.116				

**Table 9 toxics-09-00064-t009:** Output from linear mixed effects model for velocity of *A. franciscana* exposed to venlafaxine.

Comparison	F	Num df	Den df	*p*
treatment	1.180	3	116	0.321
light phase	18.062	3	394	<0.001
exposure period	3.091	2	116	0.049
treatment * light phase	0.873	9	394	0.550
treatment * exposure period	0.200	6	116	0.976
light phase * exposure period	2.095	6	394	0.053
treatment * light phase * exposure period	0.581	18	394	0.913
ICC = 0.410				
R^2^ = 0.121

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
