# Peer review of "High-Throughput Screening of Psychotropic Compounds: Impacts on Swimming Behaviours in Artemia franciscana"

_toxics, 2021, doi:10.3390/toxics9030064_

Round 1
Reviewer 1 Report
Kohler et al. manuscript demonstrates a high-throughput assay for measuring behavioural alterations in model species for ecotoxicology, the brine shrimp, Artemia franciscana. Further, Kohler et al. also use this assay to assess the impacts of serval emerging psychoactive pollutants. This is a timely and important topic. The article is well written, and the assays are well constructed. This article will be beneficial for the field—I congratulate the authors on their hard work. My main comments are about the details of the statistics section and results section. I have tried to give detailed feedback on how the authors might improve the clarity of these sections and make their data more accessible. I have run similar experiments myself, and know that the statistics can be deceptively complicated—as can be the results. I have also made a few minor suggestions throughout the MS. This was a pleasure to review.
Line 117: “21°C±1” — formatting for temperature seem a little off, maybe change to 21 ± 1 °C
Line 120: “fed 2-4 drops” — Should be an N-dash (i.e. –) not a hyphen (-) between a range of numbers.
Line 122: “1-2 days” — Same as above, and applies throughout the MS.
Figure 1: Nice photos. The boxes seem to out of alignment.
Line 147: Should have a comma before respectively.
Line 163–166: “During 2-minute dark cycles the entire arena was dark and during 2-minute light cycles one half of the arena would be illuminated and A. franciscana could choose to be in either the light or dark side of the arena.” — This is an interesting approach, like a mixture between a typical scototaxis trial and a typical phototaxis trial. It might be helpful if you explain why you chose to have only half the area illuminated during the light period. Some might argue that the 2 min dark period and 2 min light period are not comparable, as in one case the condition is consistent across the area, and in the other it is restricted to half the area. Depending on what you are actually trying to measure/compare I don’t see it as an issue.
Table 3: It’s a little unclear what plate refers to in this table. Is it just showing that you did it across two different plates?
Statistics
Line 208–209: “Phototaxis was measured as a comparison of the percent duration of time spent in dark and light zones” — It’s always a grey area to me, but would this be more akin to scototaxis then phototaxis—the definitions of the two have a significant amount of overlap but this endpoint seems to be identical to that of typical scototaxis assay?
Line 208–209: For phototaxis, is this measure only taken during the 2 min “light-periods”? Did you take the average across the two light periods? This builds on to my next point, it’s unclear how the model was structed. I assumed the model looked like this…
phototaxis ~ light.period * exposure.time.point * treatment * weight/size * total.distnace
Where:
phototaxis = response variable
light.period = the first or second light period
exposure.time.point = The first experiment (1 d) or the second experiment (7 d)
treament = the chemical treatment
weight/size = weight or size of the organism
total.distnace = total distance covered during the light photoperiod
Line 209: “velocity was measured as mean velocity in centimetres per second.” — Lot’s of my comments above also apply to the velocity endpoint. Was this scored in the dark period and light period? Was it averaged across the two light period transitions? The model specification is again unclear.
Statistics in general: It’s a little brief. I don’t think I could replicate your methods based on the information that has been provided. Can you please expand based on the points above? In addition, can you add how was the data distributed and how you checked model assumptions ect.
Figure 4. I worry that this figure could cause some level of confusion. This type of figure is typical of a photomotor response trial that has a period of complete light and complete dark, not trials like the present, which has a period of complete dark and a period with half-light half-black. I think that people may confuse the two at first glance. Maybe you can use another name for the ‘light period’, that could make it clearer.
Line 241–242: “Time also had a significant impact on phototactic response (table 5) with animals spending more time in the light zone during light phases compared to dark.” — I found this very confusing. “Time” is particularly confusing, do you mean light-period? Because technique you have both light period (dark and half-half), and time (transition 1 and transition 2). This would clearer if you explain the model structure in the statistics section.
The results you have here are more complicated than a mean effect of time, right?. You have several two-way interactions. intensity * time, size * time, and size * intensity. I would recommend you start the results section by talking about these interactions and what they mean.
Is it necessary to compare the velocity data within such fine time bins? Given you are applying a Bonferroni adjustment to each comparison, to detect a ‘significant’ effect the level of the unadjusted p-value would need to be incredibly low prior to correction, right? To avoid unnecessary comparison, and thus the need for overly harsh p-corrections, why not take the average velocity across the full 2 min time period, or even the full trial? Isn’t such a comparison just as biologically meaningful?
Reviewer 2 Report
The manuscript entitled “High-throughput screening of psychotropic compounds: im-2 pacts on swimming behaviours in Artemia franciscana” describes an interesting approach to the testing of substances in an ecotoxicological setting. The authors argue in favour of behavioural endpoints, these being particularly sensitive, and advocate the use of Brine shrimp for this purpose, because this organism has been frequently used in ecotoxicologial testing protocols and because it is easy to use in the laboratory and to procure. The authors describe the optimisation of a behavioural testing protocol with which it is possible to conduct a high number of tests (i.e. high-throughput) in a well-defined and reproducible manner. Indeed, provided the experimenter owns the system used in this work, any laboratory can reproduce the testing protocol identically, which is much less possible with behavioural tests developed in-house. The manuscript is well written and is likely to provide interesting information, even though the high-throughput tracking device as such has been described in numerous other publications. It may therefore be considered for publication in Toxics.
Nevertheless, the work presented suffers from an experimental design, which to some degree forfeits the advantages of high-throughput testing. As much as I am aware of, the experimental design contents itself with 6 replicates (N=6). For behavioural testing, this cannot be considered high-throughput and it turns out to be insufficient to prove that the observed differences in swimming velocity are statistically significant (at least for post-hoc tests).
Although behavioural tests are very sensitive, they also suffer from high inter-individual variability, which makes it difficult to obtain sufficient statistical power. For a better understanding of this variability, the authors could provide information about the coefficients of variability.
Many custom-made test protocols are labour-intensive, which limits the number of replications. The Daniovision system overcomes this problem, but the authors rather use it to test three different concentrations at three different time points, i.e. 9 different conditions each at N=6 (3x3x6 resulting in a total N of 54 plus the respective controls, is this correct?). In terms of ecotoxicology this testing scheme is certainly sensible, only it fails in producing clear and statistically significant results. As for this matter, it remains enigmatic, why the LMMs produce highly significant results for fluoxetine (as well as for the other drugs), without the post-hoc tests revealing statistically significant differences except in one case. In this respect it would be highly interesting to be able to estimate the effect sizes. Although the graphs 6 to 9 display velocity as cm·s-1 it is difficult to tell what a low and a high velocity in the heat map actually means. As far as I can judge from the left hand graphs a low velocity is situated around 0.3, whereas a high velocity is situated around 0.7. This means that the velocity differs roughly about the factor 2 (approximately a 130% from low to high). This is not much and the error bars (which are what, S.D. or S.E.M.?) are almost the same size. Given the low number of replicates with respect to the low effect size and the high variability, it is not surprising that the Tukey post-hoc test can hardly establish any significant difference. To my point of view, from these statistics the authors cannot infer that the treatment “had no effect”. The absence of statistical significance is not a proof for the absence of any effect when the combination of effect size, variability and N does not allow demonstrating statistical significance. At the least the authors should make this problem transparent when discussing their results.
Because of the supposed statistical limitations, the differences in velocity rarely fell under the p<0.05 threshold for pairwise comparisons. To enrich the manuscript, the authors describe test optimisation in vials with different diameters. Indeed, this is an interesting aspect and provides valuable information. It has, however, been reported by the same authors that the size of the arena influences behaviour. To my liking, if the paper addresses optimisation of test protocols, the authors would have done well to also carry out a power analysis. This would have constituted extremely valuable information for any researcher that would like to conduct similar tests. And it would have allowed for a better interpretation of the results, i.e. whether the differences indeed demonstrate the absence of effects or if statistical power has simply been to low. Given that the tests are simple to use and easy to repeat it is rather unfortunate that not at least some combination of promising time points and concentration has been repeated with a higher N-number.
In general, I hardly ever ask additional work for a submitted manuscript, but in this case it might be feasible that the authors add an experiment for each of the substances using only the most promising combination of time and concentration. The authors may consider if increasing the N for certain exposure conditions would enhance the chances for obtaining statistically significant results? The heat maps suggest that thismight be the case. If statistical significant differences emerge, the existing results could be considered range-finding experiments.
If the authors feel that it is not possible to conduct additional experiments (which they probably do) or if they find it unfair to ask for restarting lab-work, which is understandable, they could perhaps try to improve their statistics. To avoid false positives, the authors use Bonferroni correction, which is very conservative and usually rules out many significant differences (the authors indeed discuss this issue, but do not change the test for that matter). Another procedure, such as Benjamini-Hochberg or q-values, could perform much better. It is not too difficult to apply these procedures and hence to confirm the results from overall LMMs. Perhaps, there are other statistical approaches that can tease out statistical differences for pairwise comparisons?
In addition, I advise the authors to report precise p-values instead of cut-off p-levels. It is a conceptual mistake to consider the rejection of H1 at probability of 5% means that there is no effect, i.e. H0 ; it only means that there is none at an alpha level of 5%. But perhaps there is one at 6% or at 7%. The conventional p-value cut-offs are arbitrary. In reality alpha-levels are distributed on a continuous scale. At least one could speak of a trend in such cases. This also applies to Bonferroni cut-off values, for which one might choose 10% instead of 5%, thus turning out statistically different results with a 10% probability for type II errors (it is, however, preferable to use a different procedure for the control of type II errors). I would think that this could be perfectly acceptable. The authors could set this into a perspective for the need of increased N-numbers for such studies. In any case there is a difference between “no effect on swimming speed” and “no significant impacts on the behaviour measured”. As long as the statistical power may be questioned, it would perhaps be better to formulate precisely : “no statistically significant effect/impact could be detected”.
Whichever way, I think the results especially for fluoxetine and also for venlafaxine are coherent with what is expected and therefore point to effects of these substance on the brine shrimps. Hence, I think that the absence of statistically significant effects is rather due to a lack of statistical power, as outlined above, than to lack of effects of the chemicals. The authors should discuss this point and advise the readers in how to make better choices with regards to the study design so as to increase statistical power.
Minor comments:
Abstract – first sentence : “Animal behaviour is becoming increasingly popular … due to its increased sensitivity and speed … .”
Introduction, second phrase – Behavioural responses have been indicated as a useful endpoint as they tend to be more sensitive than lethality and faster to measure/to assess than endpoints for growth
p.2, l. 51 : the authors use both the term “model species” and “model organism”. Although other authors are speaking of a “Artemia salina as a model organism in toxicity” (Rajabi et al. (2015) DARU J Pharm Sci 23, 20) I find that this unnecessarily adds to the confusion between generally accepted “model organisms”, such as Daphnia, Drospohila, Cenorhabditis, Danio etc. and commonly used species in a specific discipline. Perhaps the term model species is less well defined and the authors may simply stick to that. Not sure whether this makes a difference. If the authors have any better idea to avoid this confusion they are welcome to use it.
p. 2, l. 77: “The MOA; presence in aquatic environments; and … “ Should not commas be used instead of semicolons?
p. 2, l. 83: Basically, I agree with the statement that vertebrates and invertebrates share several of the neurotransmitters, but in the case of epinephrine and crustaceans, I have doubts. In a 1994 review, Milton et al. (J Crust. Biol. 14: 413-437) cite several studies that detected norepinephrine (NE) in (decapod) crustaceans and report that both octopamine (OA) and NE inhibit neurons that generate the swimmeret rhythm (Mulloney et al. 1978, J. Neurophysiol. 58:584-597. Kuramoto, 1991 CBP 98A:185-190), but the existence of the respective adrenergic receptors appears to be uncertain in crustaceans (present in some protostomes [Bauknecht & Jékely, BMC Biology (2017) 15:6], but lost in insects). The authors evoke the same argument in the discussion (p. 20, “behaviour of A. franciscana as they share many of the neurotransmitters that are targeted by neuroactive drugs in vertebrates.”). I find this rather superficial and am slightly disappointed that authors do not take a more differentiated position. In some clades NE seems to be replaced by OA, whereas in others (annelids) both co-exist (but what is their respective role?). The fact that insects lost NE suggests that this might be the case in the closely related crustaceans. The above cited studies have shown that NE has lower efficiency than OA in inhibiting the swimmeret rhythm, which points to the possibility that the structurally similar NE may act on the OA receptor, but is not a good ligand. I think it would be good to reflect that there are similarities in neurotransmitters between vertebrates and invertebrates, but that there are also differences and that these differences may, in part, explain why the drugs used in this study may not exert the expected effects. This in turn, may help to explain why the tests did not reveal clear responses to produce statistically significant effects (an alternative or complementary explanation to the lack of power). Along these arguments, the authors should also give a precise account of the action of the different psychotropic drugs employed in this study.
p. 2, l. 83: typo “…, making it possible for psychotropic compounds to have effects …”
Tables 2 and 3: Why do the larger arenas have higher replicate numbers, i.e. 30 compared to 24? If this is the case, why did the authors not opt for the larger arenas, as they would have increased the N? Please explain in the Mat. & Meth.
Figure 4: The legend needs to specify what large, medium and small refers to, what is the level of significance for the asterisk and with which test it was obtained. Furthermore, the error bars have to be defined and whether the depicted values are means or medians and from how many replicates (see also comments for fig. 6-9).
Fig. 5: Similar comments as for the other figures. However, here it is stated “Asterisks indicate significant differences between arena sizes. Significance level p ≤ 0.05” (as in fig. 6).
Figures 6 to 9: The legends are insufficient. Which of the graphs relates to 1h, 1d, 1 week? One has to guess. What are the error bars? Which velocity corresponds to “high” and which corresponds to “low”? For the significance level, what was the test and what was the N?
Reviewer 3 Report
This is an interesting study that deserves to be published but after implementing with the following suggestions. The authors perform the study with adults, which are quite large compared with nauplii and Daphnia juveniles. This means that it is expected that for large animals larger arenas should be better suited. Authors should provide the mean size of body length of the tested organisms to allow readers comparison with other organisms. I liked a lot the phototactic device of using acrylic fibres. It is a shame that this set up was not used for neuroactive drugs.
There are too many Figures (9) and 4 of them (from 6 to 9)show similar data structure from, which it is difficult to obtain clear results. I suggest to add them to SI and make a new figure with the mean values like Fig 4A with the velocity results of all neuro-active compounds tested.
About the discussion. It is nice but there are few issues that need to be addressed. First it deserves more discussion the observed no differences between 100 and 5% light intensity. Secondly the observed higher speed under dark contrast to what it has been reported for other crustacean but it is an interesting feature. It is unlikely for Artemia to have a diel vertical migration associated to predation since it lives in hypersaline environments where fish is often absent. Anyway the authors are probably right that this can be related to the fact that under darkness animals increase search for light. In fishless habitats having a positive phototactic behaviour may be favourable since edible algae may be more abundant in the surface of hypersaline lakes. About the observed decrease of speed in the microplates across exposure times it is likely to be related to suboptimal growth conditions. For Daphnia, which is smaller in size of Artemia it is recommended a volume of 50-100 mL in reproduction tests. The proposed habituation is an issue but it occurs to repetitive stimuli separated seconds or min, no days or weeks
Round 2
Reviewer 2 Report
The authors have considerably changed their manuscript and provided extended information. I find the manuscript much improved.
It is true that the number of replicates was mentioned in a table (and actually I asked a question about it), but I did not understand how the replicates were composed. This information is now provided in the figure legend to figure 2. I would think that this information should also be included in text to the materials and methods. To my point of view, it should be possible to read text and figure legends independently.
Given that the replications were at least 24, indeed a lack of statistical power is unlikely. If the results show no statistically significant difference, then this is certainly due to a absence of a clear response. Overall, the study is indead high-throughput and a such the results are highly interesting.
The authors have answered all my questions and I have no further comments. I would only advise to include the information on the replicates now found in the legend to figure 2 in the methods text.